# The Impact of COVID-19 on Women’s Physical Activity Behavior and Mental Well-Being

**DOI:** 10.3390/ijerph17239036

**Published:** 2020-12-04

**Authors:** Carl P. Nienhuis, Iris A. Lesser

**Affiliations:** Faculty of Health Sciences, Kinesiology, University of the Fraser Valley, 45190 Caen Ave, Chilliwack, BC V2R 0N3, Canada; carl.nienhuis@ufv.ca

**Keywords:** 2019 novel coronavirus diseases (COVID-19), women’s health, physical activity, well-being, anxiety

## Abstract

Background: A global pandemic caused by COVID-19 resulted in restrictions to daily living for Canadians, including social distancing and closure of recreation facilities and provincial parks. Methods: The objective of this study was to assess whether sex differences exist in physical activity and well-being since COVID-19 and to explore how barriers or facilitators to physical activity may explain these differences. Chi-square tests, independent t-tests and one-way ANOVAs were conducted to evaluate data provided by 1098 Canadians—215 men and 871 women. Results: Women were significantly less physically active than men and reported more barriers and fewer facilitators to physical activity and experienced significantly more generalized anxiety than men. Women who were engaged in less physical activity due to COVID-19 reported significantly lower mental health scores, lower social, emotional and psychological well-being, and significantly higher generalized anxiety, while women who engaged in more physical activity had improved mental health scores. Conclusions: Given the challenges that women uniquely face due to restrictions, it is imperative to advocate and provide environmental opportunity and support for physical activity to reduce the mental duress women may be experiencing. Specific physical activity programming that is inclusive of lifestyle physical activity and can engage children is encouraged.

## 1. Introduction

A global pandemic was declared in March of 2020 due to coronavirus 19 (COVID-19), a potentially fatal respiratory virus, with the first cases reported in Wuhan, China. As well as the direct impact of the disease, there may be unintended negative health consequences due to public health restrictions such as reduced health-promoting behaviors and increased mental duress [1,2,3,4] due to social confinement [4,5]. Additionally, women may be disproportionately affected by the negative consequences of COVID-19 restrictions as they make up 70% of health and social-service workers worldwide [6] and are more likely to be in retail and service jobs. Recent statistics in Canada have shown that women between the ages of 25 and 54 had twice the rate of job loss as seen by men since the COVID-19 pandemic was declared [7]. Little is known about the effect of COVID-19 and associated consequences on Canadian women’s mental health and overall well-being.

The ability to engage in physical activity can mediate the stress response and promote well-being. For instance, every session of physical activity has been shown to positively impact symptoms of depression and anxiety [8] and therefore can serve as an acute coping mechanism with additional positive benefits on physical health. Women experience higher levels of anxiety disorders [9] and experience more generalized anxiety than men [10]. Given that women tend to be less physically active than men [11,12] and the prevalence of insufficient physical activity is growing amongst women in high-income countries [12], the consequences of the COVID-19 restrictions may result in further mental health disparity between men and women.

A reduction in health-promoting behaviors due to preventive public health measures such as social distancing and closure of recreation centers, city parks and playgrounds would have a potential impact on physical activity engagement. Physical activity engagement is a multi-faceted construct impacted by individual factors such as motivation and exercise self-efficacy, as well as environmental factors such as social support and recreational opportunity. Women have reported a greater number of barriers to exercise than men and these were associated with lower physical activity participation in pre pandemic studies [13]. Lack of enjoyment, self-consciousness and time constraints are frequently cited by women as barriers to being physically active [14]. Individual and environmental factors have the potential to be affected by the public health restrictions and impact men and women differently. Additionally, preventive public health measures altered childcare responsibilities with childcare centers and schools closing. Women with young children have routinely been shown to have lower physical activity engagement than those without [15]. Increased responsibility at home could impact health-promoting behaviors and mental health, with women potentially carrying a greater amount of this responsibility.

The primary objectives of this study were to assess whether sex differences exist in physical activity and well-being since COVID-19 and to explore how barriers or facilitators to physical activity may explain these differences. As a secondary objective, we aimed to look at whether changes in occupation or childcare responsibilities since COVID-19 have impacted women’s well-being and physical activity engagement.

## 2. Materials and Methods

### 2.1. Sample

Study participants were Canadian men (*n* = 215) and women (*n* = 871) over the age of 19 who were recruited for a national study on physical activity and well-being during COVID-19. The original study has been described in detail elsewhere [16] and had a cross-sectional study design. Study participants were recruited through regular media communications including stories in national and local media and snowball sampling using social media. This study received approval from the Human Research Ethics Board at the University of the Fraser Valley (100449) and all participants provided online informed consent and were ensured of anonymous data collection.

### 2.2. Measures

Questionnaires were completed by participants in April and early May 2020 during the strictest public health restrictions in Canada using an online survey software (Survey Monkey). The survey included demographics, physical activity behavior, and measures of well-being. Demographic characteristics included age, sex, marital status, occupational status (including changes due to COVID-19) and changes in childcare obligations (due to COVID-19).

#### 2.2.1. Physical Activity Behavior

Participants reported their current physical activity levels using the Godin Leisure Questionnaire [17] at the time of survey completion during COVID-19. To determine whether participants were physically active, amounts of reported vigorous and moderate physical activity participation in the Godin questionnaire were used to categorize participants as active (>150 min of moderate–vigorous physical activity per week) or inactive. This was previously utilized as a method to categorize physical activity in studies including women [18]. Participants were also requested to indicate whether physical activity had changed (same, more or less) since COVID-19 restriction onset. The perceived benefits, enjoyment, confidence, difficulty and planning in physical activity behavior were assessed to determine barriers and facilitators to physical activity. Additional questions were asked to evaluate the potential impact of social distancing on challenges, support and opportunity for engagement in physical activity.

#### 2.2.2. Motivation

Participant motivation to exercise was assessed at the time of survey completion utilizing the Behavioral Regulations in Exercise Questionnaire (BREQ-3), an instrument that has been shown to be valid and reliable in both women and men [19,20,21,22]. The BREQ-3 measures amotivation (e.g., lacking intent to exercise), external regulation (e.g., exercising because one tells you to), introjected regulation (e.g., feeling guilty when one does not exercise), identified regulation (e.g., valuing the benefits of exercise), integrated regulation (e.g., exercise is part of personal identity), and intrinsic motivation (e.g., exercising because one enjoys it) [23] and further aggregate scores of controlled and autonomous motivation are given [24].

#### 2.2.3. Anxiety

The General Anxiety Disorder-7 (GAD-7) was used to identify participant anxiety at the time of survey completion during COVID-19 [25]. The validity of GAD-7 was substantiated in a large primary care sample of men and women with a sensitivity value of 0.89 and a specificity value of 0.82 [26]. The GAD-7 is based on seven items that are scored on a scale of 0 to 3 with a total possible score of 21. Cut off scores of 5, 10 and 15 were used as a score of mild-, moderate and- severe-anxiety symptoms, respectively [27].

#### 2.2.4. Well-Being

Participant emotional, psychological and social well-being was assessed at the time of survey completion during COVID-19 using the Mental Health Continuum Short Form (MHC-SF) [28], a scale demonstrated to have strong internal consistency and test–retest reliability [29]. In addition to providing an overall well-being score, the scale was used to classify participants as flourishing or languishing for further data analysis.

### 2.3. Statistical Analysis

Descriptive statistics of demographic characteristics were conducted, and independent *t*-tests and chi-square tests were conducted to compare demographic differences across sex. To analyze physical activity behavior and well-being outcomes of women, participants were categorized based on changes to physical activity behavior and subsequent comparative analysis was performed utilizing one-way ANOVA. Multiple one-way ANOVAs were conducted to examine the number of moderate–vigorous physical activity minutes and barriers and facilitators to physical activity engagement. Bivariate analysis and independent sample t-tests were conducted to compare motivation levels between men and women, and additional t-tests were utilized to explore the impact of childcare changes on physical activity and well-being measures. SPSS-25.0 software was utilized to compute all statistical analysis and significance was set at *p* < 0.05.

## 3. Results

Overall, 1098 responses were collected, including 215 men (mean age 45 ± 16) and 871 women (mean age 41 ± 15). Initial descriptive analysis and chi-square tests explored differences between women and men (see Table 1). Among women, 65.9% reported being married or in a domestic relationship, 24.9% were single, and the remaining 9.1% were either widowed, divorced or separated. There were significant differences in the sample regarding education levels between men and women; *X*^2^(12, *n* = 1098) = 24.68, *p* = 0.016. Specifically, a greater portion of women (27.7%) completed graduate school compared to men (20.0%). Regarding employment status prior to COVID-19, 55.7% of women were employed full time compared to 67% of men, and 21.7% of women were employed part time compared to 6% of men. These frequencies in employment status were significantly different between men and women; *X*^2^(15, *n* = 1092) = 57.48, *p* < 0.001. Further, there was a significant difference between men and women and change in work; *X*^2^(3, *n* = 1098) = 25.88, *p* < 0.001. While only 43% of men experienced changes to their work on account of COVID-19, 60% of women experienced work-related changes, including reduced hours (10.7%), remote work (32.8%) or loss of employment (16.2%). Finally, there was no significant difference between men and women who saw increased demands to care for children that would otherwise be in school or childcare; *X*^2^(3, *n* = 1098) = 2.39, *p* = 0.496.

Table 2 displays the results of independent sample t-tests which explored differences between men and women and various physical activity and well-being measures. Women were significantly less physically active than men, and women experienced significantly more generalized anxiety than men. Active women (M = 49.60, SD = 11.66) scored significantly higher (*p* = 0.045) on the Mental Health Continuum than inactive women (M = 47.82, SD = 12.89), though there was a non-significant difference between activity levels on generalized anxiety (*p* = 0.455). From the sample, 34.8% of women were categorized with mild anxiety, 36.7% with moderate anxiety, and 17.2% with severe anxiety. There was a significant difference between levels of anxiety and minutes of moderate–vigorous physical activity [F(3,849) = 3.24, *p* = 0.022] in women. Specifically, women with severe anxiety reported more physical activity than women with moderate anxiety (M_diff_ = 44.37, SE = 15.60, *p* = 0.013). No significant differences between men and women were found on the Mental Health Continuum.

Table 3 reports ANOVA findings that evaluated how changes in physical activity since COVID-19 related to well-being and anxiety in women. Overall, 37.3% of women became more active, 28.1% maintained similar activity levels, and 34.6% became less active. There was a significant difference between physical activity changes on all measures of mental well-being, including social, emotional and psychological well-being, as well as generalized anxiety (Table 2). Tukey post hoc tests revealed that women who were engaged in less physical activity due to COVID-19 reported significantly lower Mental Health Continuum scores, lower social, emotional and psychological well-being, and significantly higher generalized anxiety.

While 35.4% of the women in the sample engaged in similar types of physical activity as prior to COVID-19 restrictions, 25.9% were not able to. Whether women were able to continue to participate in similar physical activity had a significant effect on overall moderate–vigorous physical activity [F(2866) = 6.18, *p* = 0.002] and generalized anxiety [F(2848) = 4.95, *p* = 0.007]. Women not engaged in similar activities were significantly more anxious (M_diff_ = 1.23, SE = 0.40, *p* = 0.005) and less active (M_diff_ = 48.40, SE = 13.77, *p* = 0.001) than those who engaged in similar activities.

Barriers and facilitators to physical activity in relation to amount of moderate–vigorous physical activity engagement is shown in Table 4. There was a significant difference (*p* < 0.001) in the number of minutes of moderate–vigorous physical activity based on the presence or absence of all measured barriers and facilitators. While men also indicated an impact on physical activity with greater physical activity associated with physical activity facilitators, the barriers to physical activity had a non-significant effect on physical activity levels (see Table 5). That is, women’s physical activity levels were more significantly impacted by the increased difficultly and challenge due to the onset of COVID-19 restrictions. Specifically, women that reported more extreme difficulty and challenge in being active due to COVID-19 reported significantly less moderate–vigorous physical activity than women who viewed these barriers as less impactful. Similarly, women that reported physical activity as more enjoyable and beneficial also reported significantly more minutes of physical activity than women who did not view physical activity as enjoyable or beneficial. Women who reported less planning and confidence in physical activity engagement participated in significantly less physical activity than women with more confidence and more detailed physical activity plans. Lastly, women that indicated experiencing less opportunity and less social support for physical activity were significantly less active than women who received a moderate or high amount of opportunity and support for physical activity.

Analysis was conducted to explore differences between men and women according to motivation and physical activity. Of the six types of behavioral regulation, women reported significantly higher introjected regulation scores than men (M_diff_ = 0.23, SE = 0.08, *p* = 0.005). Bivariate analysis indicated significant (*p* < 0.05) and moderately strong correlations between autonomous motivation and the number of minutes that women spent in moderate–vigorous physical activity (*r* = 0.42). Further tests explored motivational differences between women based on changes in physical activity due to COVID-19. Results showed significant between group differences for autonomous motivation [F(2868) = 12.61, *p* < 0.001]. Post hoc tests demonstrated that women that maintained the same level of activity reported significantly more autonomous motivation than those that were more active (M_diff_ = 0.25, SE = 0.07, *p* = 0.001) or less active (M_diff_ = 0.36, SE = 0.07, *p* < 0.001).

Analysis explored the impact of changes in childcare provision with women on levels of physical activity and well-being measures (see Table 6). While there were no between-group differences regarding levels of physical activity, women responsible for increased childcare provision reported significantly (*p* = 0.004) more generalized anxiety than women who saw no changes in childcare. Further, emotional well-being significantly (*p* = 0.010) differed between women based on childcare demands.

Subsequent analysis evaluated differences between less active and more active women and well-being outcomes. Inactive women who were responsible for increased childcare reported higher levels of generalized anxiety (M_diff_ = 1.49, SE = 0.44, *p* = 0.001) and higher levels of emotional well-being (M_diff_ = 0.57, SE = 0.27, *p* = 0.036), while the active participants did not show a significant change in anxiety or well-being levels on account of childcare changes. Regarding barriers and facilitators of physical activity, women who experienced increased childcare demands indicated increased difficulty (M_diff_ = 0.64, SE = 0.10, *p* = 0.006) and decreased confidence (M_diff_ = 0.21, SE = 0.09, *p* = 0.025) in being physically active. As well, women without changes in childcare provision reported more opportunity to be physically active (M_diff_ = 0.36, SE = 0.08, *p* < 0.001).

While work changes had no significant impact on levels of physical activity, an independent samples t-test revealed that women who experienced work-related changes reported significantly more generalized anxiety than women who experienced no changes in their work due to COVID-19 (M_diff_ = 0.90, SE = 0.32, *p* = 0.006). That is, women who lost their jobs, began working remotely, or working fewer hours were more anxious then women who maintained regular work routines. As well, motivation differed in women on the basis of work-related changes. Women who reported changes in their work also reported significantly more introjected regulation (M_diff_ = 0.21, SE = 0.07, *p* = 0.004) and controlled motivation (M_diff_ = 0.12, SE = 0.05, *p* = 0.014) than women who did not report changes in their work. In other words, women who experienced work-related changes were more motivated to engage in physical activity on the basis of external rewards, punishment or feelings of guilt and pressure than women who reported no changes in work routines. These results indicate that women who experienced work-related changes experienced less self-determined motivation to engage in physical activity.

## 4. Discussion

Women were significantly less physically active than men and reported more barriers and fewer facilitators to physical activity. Women experienced significantly more generalized anxiety than men and women with changes to work or childcare provision due to COVID-19 were more anxious. Women who were engaged in less physical activity due to COVID-19 reported significantly lower mental health scores, lower social, emotional and psychological well-being, and significantly higher generalized anxiety, while active women had improved mental health scores.

Women engaged in less physical activity than men during public health restrictions which parallels what has previously been shown regarding sex differences in physical activity [12]. Overall, 37.3% of women became more active, 28.1% maintained similar activity levels, and 34.6% became less active since the onset of COVID-19 restrictions. Perceived or real physical activity barriers directly affect physical activity participation. We found that there were significant differences between barriers and facilitators to physical activity and the amount of moderate–vigorous physical activity. Women were specifically impacted by barriers of increased difficultly and challenge in engaging in physical activity during public health restrictions. Our results parallel what is seen outside of the COVID-19 context with less active women more likely to report barriers to physical activity including fatigue and lack of interest in physical activity than physically active women [30]. Women who reported lower levels of physical activity have expected or perceived physical activity to not be enjoyable, while physically active women commonly report physical activity to be enjoyable [31]. Additionally, self-confidence is more strongly associated with physical activity engagement in women due to concern about body size, shape and athletic ability when performing physical activity [32]. Lastly, lack of time is a common barrier to physical activity among working women [31] and even more so amongst women with children under the age of 15 [33] with lack of childcare cited as a reason for physical inactivity [34]. With the public health restrictions closing childcare centers and schools a lack of childcare would be of increased challenge in physical activity engagement for women. Women who reported having increased childcare demands indicated increased difficulty and decreased confidence in being physically active. As well, women without changes in childcare provision reported more opportunity to be physically active.

We found that women who maintained their physical activity since COVID restrictions were more autonomously motivated and had greater autonomous motivation related to increased levels of physical activity. Women also reported more introjected regulation than men which involves pursuing an activity due to feelings of pressure or compulsion [35]. We also found that women who reported changes in their work reported significantly more introjected regulation and controlled motivation. According to self-determination theory and substantiated by research, while introjected regulation can facilitate behavioral compliance and accompanying positive affect, the controlling dimensions of introjected regulation can lead to increased feelings of anxiety and decreased confidence and well-being [36]. Introjected regulation is associated with short-term behavioral engagement but not long-term behavioral persistence [37]. Though internal pressures or avoidance of guilt and shame can initiate early behavioral change, our findings are consistent with the self-determination continuum in that women motivated by more autonomous regulatory behavior experienced improved psychosocial outcomes, including a reduction in generalized anxiety [38,39].

Women who reported lower levels of physical activity due to COVID-19 reported significantly lower Mental Health Continuum scores, lower social, emotional and psychological well-being, and significantly higher generalized anxiety. While a large amount of research has been conducted on the positive impacts on anxiety symptomology through physical activity engagement in adults, less have looked specifically at women [40,41]. A Norwegian study found women to be twice as likely to have anxiety symptomology than men, and women who had higher scores in moderate–vigorous physical activity had lower levels of anxiety symptomology [42]. An Irish study in older adults found the symptoms of worry in women were greater amongst those that did not meet physical activity guidelines, but the magnitude of difference was small [41]. Interestingly, a study in Belgium found differing physical activity impacts on anxiety in men and women with women reporting improved anxiety symptomology with moderate-intensity exercise and improved emotional well-being with walking, while men required moderate-intensity exercise and exercise primarily addressed physical complaints of anxiety [43]. While it appears that physical activity has a positive impact on anxiety in men and women it is possible that the degree of this impact may be different in women. This parallels a recent study out of Italy which found the correlation between physical activity and psychological well-being to be higher in females than males suggesting that variations in physical activity since COVID-19 had greater influence on the psychological status of women than men [44].

Additionally, women who reported altering their physical activity type since COVID-19 restrictions were significantly more anxious and less active than those that maintained the physical activity type they engaged in prior to COVID-19. A lack of confidence or ability to engage in novel physical activity options may further increase anxiety among women. Lower self-efficacy for exercise has been associated with lower physical activity enjoyment [45] and depressive symptomology in women [46]. Interestingly, when categorizing anxiety, we found that women who rated as having severe anxiety were more physically active than those who had moderate anxiety. It is possible that women who had more severe anxiety were more avid exercisers before the onset of COVID-19 and may have used physical activity as a coping mechanism with the additional mental duress [47]. However, women who were physically active did have better mental health scores than those who were inactive.

Mean scores on the generalized anxiety scale were significantly higher in women than men in our sample with 34.8% of women reporting mild anxiety, 36.7% moderate anxiety, and 17.2% severe anxiety. The GAD-7 has been shown to be strongly correlated with mental health in a sample of patients assessed in primary care [26]. A score of 10 on the GAD-7 has been suggested to be a reasonable cut point for identifying cases of generalized anxiety [26]; in our general voluntary population sample the average GAD-7 score in women was 10 with just over half of women having moderate to severe anxiety. It is unknown whether this concerningly high anxiety score in our population of women would have been found prior to COVID-19.

Women have been additionally affected by the economic hardships and increased childcare responsibilities. We found that women with childcare responsibilities due to COVID-19 had higher anxiety than those who did not have additional childcare demands. Women who experienced increased childcare demands reported increased difficulty and decreased confidence in being physically active. As well, inactive women who were responsible for increased childcare reported higher levels of generalized anxiety, while the active participants did not show a significant change in anxiety levels on account of childcare changes. These findings suggest that women who are able to overcome the barriers associated with childcare demands are able to reduce their anxiety through physical activity. While we found no significant difference between men and women who saw changes to childcare provision, the question did not ask who was bearing the brunt of childcare responsibility, including homeschooling. Women with childcare responsibility have often been shown to impair career productivity in medical professions [48], while women within academics have been shown to have reduced productivity since the onset of COVID-19 [49]. Relatedly, we found that while 43% of men experienced changes to their work on account of COVID-19, 60% of women experienced work-related changes, including reduced hours (10.7%), remote work (32.8%) or were laid off (16.2%). Work-related changes would likely place women at increased economic hardship with associated mental duress [3]. We found that women who were laid off, working remotely, or working fewer hours were more anxious than women who maintained regular work routines.

Recruitment methods that utilized a web-based voluntary approach would result in some selection bias. It is likely that this sampling methodology recruited a population of women with a higher level of education with over a quarter of the sample having graduate education. Higher education has been associated with greater physical activity engagement [50]. Further, assessing well-being, anxiety and activity levels prior to COVID-19 restrictions would have enabled a more accurate evaluation of the impact of COVID-19 on these measures. Lastly, a more robust sampling of childcare demands would have provided a better depiction of the impacts of the pandemic on parenting and associated physical activity and mental well-being in women.

## 5. Conclusions

Given the challenges that women uniquely faced during COVID-19 restrictions, it is imperative to advocate and provide environmental opportunity and support for physical activity to reduce the mental duress women may be experiencing. Specific physical activity programming that is inclusive of lifestyle physical activity and can engage children or provide support and opportunity for physical activity engagement is encouraged. Additionally, the use of digital technology may reduce the psychosocial strain of home confinement [4,5] by providing opportunity to engage in socially supportive physical activity. Future research should assess the long-term impact of the pandemic on women’s physical activity and associated mental well-being as well objective measurement of physical activity.

## Figures and Tables

**Table 1 ijerph-17-09036-t001:** Participant demographics split by sex.

Participant Characteristics	Male	Female	Total	*p*-Value
	*N* (%)	*N* (%)	*N* (%)	
	215 (19.6)	871 (79.3)	1098 (100)	
Age (mean, SD)	45 ± 16	41 ± 15	42 ± 15	<0.001
Relationship Status				0.015
Married/Domestic	168 (78.1)	574 (65.9)	752 (68.5)	
Widowed/Divorced/Separated	8 (3.7)	79 (9.1)	88 (8.0)	
Single	39 (18.1)	217 (24.9)	257 (23.4)	
Age				0.095
Under 20	1 (0.5)	9 (1.0)	10 (0.9)	
20–29	39 (18.1)	220 (25.3)	261 (23.8)	
30–39	57 (26.5)	243 (27.8)	304 (27.7)	
40–49	34 (15.8)	151 (17.3)	191 (17.4)	
50–59	31 (14.4)	100 (11.5)	131 (11.9)	
60–69	36 (16.7)	102 (11.7)	138 (12.6)	
70+	14 (6.5)	34 (3.9)	48 (4.4)	
Employment status (pre COVID)				<0.001
Full time	144 (67)	485 (55.7)	640 (58.3)	
Part time	13 (6)	189 (21.7)	203 (18.5)	
Unemployed	10 (4.7)	41 (4.7)	51 (4.6)	
Homemaker	1 (0.5)	42 (4.8)	43 (3.9)	
Retired	46 (21.4)	99 (1.4)	145 (13.2)	
Unable to work	0 (0)	10 (1.1)	10 (0.9)	
Employment status (post COVID)				<0.001
No change	122 (56.7)	350 (40.2)	473 (43.2)	
Reduced hours	17 (7.9)	93 (10.7)	110 (10)	
Remote work	58 (270)	286 (32.8)	352 (32.1)	
Laid off	17 (7.9)	141 (16.2)	161 (14.7)	
Childcare				0.502
Yes	51 (23.7)	224 (25.7)	278 (25.3)	
No	164 (76.3)	646 (74.2)	819 (74.7)	

**Table 2 ijerph-17-09036-t002:** *t*-Tests Results Comparing Women and Men on Physical Activity and Well-Being Measures.

	Women	Men	*p*
	*N* = 871	*N* = 215	
	M ± SD	M ± SD	
Physical Activity Measures			
Godin Leisure Score	143.43 ± 123.21	187.56 ± 187.21	<0.001
Moderate–Vigorous Physical Activity *	140.40 ± 158.00	183.50 ± 191.09	0.001
All Physical Activity *	416.57 ± 332.32	539.48 ± 591.88	<0.001
Well-Being Measures			
GAD-7	10.40 ± 4.63	8.74 ± 4.63	<0.001
MHC Score	48.44 ± 12.49	48.07 ± 11.67	0.706
Social Well-Being	15.13 ± 5.64	14.73 ± 5.50	0.359
Emotional Well-Being	11.50 ± 2.73	11.79 ± 2.80	0.174
Psychological Well-Being	21.76 ± 5.79	21.47 ± 5.36	0.513

Notes. * Moderate–vigorous physical activity and all physical activity expressed as min/week.

**Table 3 ijerph-17-09036-t003:** ANOVA results for well-being outcomes in relation to changes in physical activity since COVID-19.

Well-Being Scales	Women			
	*N* = 871			
Change in Physical Activity Since COVID-19	More Active	Same	Less Active	*p*
	*N* = 325	*N* = 245	*N* = 301	
	M ± SD	M ± SD	M ± SD	
Mental Health Continuum Score	49.63 ± 12.18	50.70 ± 10.99	45.36 ± 13.37	<0.001
Social	15.75 ± 5.60	15.81 ± 5.13	13.90 ± 5.89	<0.001
Emotional	11.77 ± 2.52	12.07 ± 2.31	10.75 ± 3.09	<0.001
Psychological	22.14 ± 5.58	22.79 ± 5.32	20.55 ± 6.16	<0.001
GAD -7	9.96 ± 4.41	10.00 ± 4.60	11.20 ± 4.78	0.001

**Table 4 ijerph-17-09036-t004:** ANOVA results for barriers and facilitators to physical activity for women and number of minutes of moderate–vigorous physical activity (min/week).

	Women			
	*N* = 871			
	Low	Moderate	High	*p*
	M ± SD	M ± SD	M ± SD	
Barriers				
How difficult is PA right now?	161.62 ± 152.23	141.63 ± 166.95	90.16 ± 150.91	<0.001
Is PA more challenging now?	155.55 ± 160.11	166.76 ± 144.35	106.12 ± 155.38	<0.001
Facilitators				
How planned is your PA right now?	84.23 ± 122.97	132.49 ± 150.03	216.60 ± 171.49	<0.001
How many opportunities do you have for PA?	82.87 ± 174.86	133.38 ± 156.35	160.62 ± 150.03	<0.001
How beneficial is PA right now?	29.85 ± 48.31	75.05 ± 203.38	149.35 ± 154.42	<0.001
How enjoyable is PA right now?	58.10 ± 82.47	91.55 ± 95.54	175.04 ± 174.80	<0.001
How confident are you to be PA right now?	53.87 ± 67.04	113.01 ± 156.89	173.49 ± 165.30	<0.001
Do you have support to be active right now?	100.28 ± 160.13	140.69 ± 172.12	165.31 ± 143.05	<0.001

**Table 5 ijerph-17-09036-t005:** ANOVA results for barriers and facilitators to physical activity for men and number of minutes of moderate–vigorous physical activity (min/week).

	Men			
	*N* = 215			
	Low	Moderate	High	*p*
	M ± SD	M ± SD	M ± SD	
Barriers				
How difficult is PA right now?	208.18 ± 190.08	157.46 ± 160.08	149.39 ± 216.83	0.111
Is PA more challenging now?	190.31 ± 173.51	197.64 ± 201.98	162.34 ± 218.19	0.584
Facilitators				
How planned is your PA right now?	130.85 ± 184.07	211.93 ± 197.62	225.06 ± 182.21	0.003
How many opportunities do you have for PA?	133.75 ± 239.43	144.64 ± 130.70	213.82 ± 197.13	0.021
How beneficial is PA right now?	24.33 ± 23.16	148.24 ± 242.97	191.59 ± 187.03	0.078
How enjoyable is PA right now?	80.78 ± 162.29	124.25 ± 164.99	221.55 ± 194.05	<0.001
How confident are you to be PA right now?	136.21 ± 224.49	122.42 ± 142.20	209.93 ± 187.52	0.014
Do you have support to be active right now?	149.23 ± 173.31	135.05 ± 114.41	236.56 ± 225.44	0.001

**Table 6 ijerph-17-09036-t006:** *t*-Tests Results Comparing Women Responsible for Increased Childcare Provision and Women with No Changes to Childcare Provision.

	Childcare Change	No Childcare Change	*p*-Value
	*N* = 225	*N* = 646	
	M ± SD	M ± SD	
Physical Activity Measures			
Godin Leisure Score	142.74 ± 141.25	143.80 ± 116.47	0.912
Moderate–Vigorous Physical Activity *	140.61 ± 179.54	140.47 ± 150.04	0.991
All Physical Activity *	412.58 ± 377.63	418.65 ± 315.51	0.823
Well-Being Measures			
GAD-7	11.17 ± 4.64	10.13 ± 4.59	0.004
MHC Score	49.72 ± 12.07	48.00 ± 12.62	0.084
Social Well-Being	15.72 ± 5.48	14.92 ± 5.69	0.076
Emotional Well-Being	11.91 ± 2.49	11.36 ± 2.80	0.01
Psychological Well-Being	22.00 ± 5.84	21.67 ± 5.78	0.475

Notes. * Moderate–vigorous physical activity and all physical activity expressed as min/week.

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
