# Peer review of "The Impact of COVID-19 on Women’s Physical Activity Behavior and Mental Well-Being"

_ijerph, 2020, doi:10.3390/ijerph17239036_

Round 1

Reviewer 1 Report

Nienhuis & Lesser report in their manuscript very well the effects of COVID-19 on Women’s Physical Activity Behavior and Mental Well-being. I have only some references from the the ECLB-COVID19 consortium i would include in the manuscript:

  • DOI:10.1371/journal.pone.0240204
  • DOI:10.5114/biolsport.2021.100149
  • DOI:10.3390/ijerph17176237

Furthermore, it would be nice if the authors could expand in the discussion a bit more what the next research steps should be regarding this research topic (long term effects of COVID19 on physical activity and health). Discuss also open questions and possible new experimental ideas a bit more deeply (e.g. possibilities to acces physical activity levels).

Author Response

Nienhuis & Lesser report in their manuscript very well the effects of COVID-19 on Women’s Physical Activity Behavior and Mental Well-being. I have only some references from the the ECLB-COVID19 consortium i would include in the manuscript:

Congratulations on these papers and thank you for letting us know to include this in our manuscript.

The following has been added to the conclusion. “Additionally, the use of digital technology may reduce the psychosocial strain of home confinement (Ammar et al., 2020 (DOI:10.5114/biolsport.2021.100149, DOI:10.3390/ijerph17176237) by providing opportunity to engage with other women and social support for physical activity.”

Additionally, references (DOI:10.1371/journal.pone.0240204 and DOI:10.3390/ijerph17176237) have been added in the introduction to support the objectives of this paper.

Furthermore, it would be nice if the authors could expand in the discussion a bit more what the next research steps should be regarding this research topic (long term effects of COVID19 on physical activity and health). Discuss also open questions and possible new experimental ideas a bit more deeply (e.g. possibilities to acces physical activity levels).

The following has been added to the conclusion based on your comments.

“Additionally, the use of digital technology may reduce the psychosocial strain of home confinement (Ammar et al., 2020 (1 and 2)) by providing opportunity to engage in socially supportive physical activity. Future research should assess the long term impact of the pandemic on women’s physical activity and associated mental well-being as well objective measurement of physical activity.”

Reviewer 2 Report

Thank for writing an interesting article on an important subject now that we are beginning to see effects from the pandemic. I have here some overall comments and more in the attached document.

Overall, I see it as a major problem that you refer to unpublished data without explanation. It is impossible to review if the data is hidden.  

It is very interesting that you write that specific physical activity programming that is inclusive of lifestyle physical activity and can engage children is encouraged, since many gyms and workout sessions that were cancelled instead offered digital training/fitness program that also engaged children. However, it would also be interesting if you also look at or only discuss how countries without lockdown tackled the challenge of physical activity. 

in other countries the potential other public health consequences have been less, since avoiding strict lockdown and allowing health promoting activities (and also keeping schools open, especially free-schools). Also providing generous social security and unemployment benefits can avoid negative health effects.  

In the introduction you write that women tend to be less active than men and use a reference from the US. But is this a general notion? In countries like Sweden women are more physical active than men? Maybe a question of equality question? Could be beneficial to use to international data to broaden the discussion? You use a reference from 2001 to show that women with young children are less physical active, please update this statement with newer international data. 

When describing your sample you refer to an unpublished study which makes it hard to check so can you describe your sample? This goes also for many parts of your results that need to be backed with your research data, either as supplement or as an open source data base. 

Author Response

Thank for writing an interesting article on an important subject now that we are beginning to see effects from the pandemic. I have here some overall comments and more in the attached document.

Thank you for taking the time to read our article and provide specific comments to improve it.

Overall, I see it as a major problem that you refer to unpublished data without explanation. It is impossible to review if the data is hidden.  

Thank you for noting this. We apologize that we missed updating this earlier with the appropriate published article. This has now been updated in the reference section.

It is very interesting that you write that specific physical activity programming that is inclusive of lifestyle physical activity and can engage children is encouraged, since many gyms and workout sessions that were cancelled instead offered digital training/fitness program that also engaged children. However, it would also be interesting if you also look at or only discuss how countries without lockdown tackled the challenge of physical activity. 

Thank you for this comment. The following phrase was included in the conclusion which recognizes that alternative mechanisms were utilized to maintain physical activity adherence:

“Additionally, the use of digital technology may reduce the psychosocial strain of home confinement (Ammar et al., 2020) by providing opportunity to engage in socially supportive physical activity.”

While the objectives of this paper were specific to human behavior in a lockdown environment it would be interesting to assess differences between lockdown and restrictions on physical activity behavior and mental health in a secondary study.

in other countries the potential other public health consequences have been less, since avoiding strict lockdown and allowing health promoting activities (and also keeping schools open, especially free-schools). Also providing generous social security and unemployment benefits can avoid negative health effects.  

We agree that this data may be specific to Canadian women given how each country had handled restrictions and employment benefits. Given that this data was retrieved at the most extensive phase of lockdown in Canada it is representative of that state of restriction. We also acknowledge that this was a cross-sectional analysis and therefore lacks the longitudinal design that would determine if mental health and physical activity continued to be altered as restrictions changed and the length of the pandemic increased.

In the introduction you write that women tend to be less active than men and use a reference from the US. But is this a general notion? In countries like Sweden women are more physical active than men? Maybe a question of equality question? Could be beneficial to use to international data to broaden the discussion? You use a reference from 2001 to show that women with young children are less physical active, please update this statement with newer international data. 

Thank you for this. We agree that this could be improved and have added a reference based on international analysis published in the Lancet which shows that women have lower physical activity levels than men worldwide.

When describing your sample you refer to an unpublished study which makes it hard to check so can you describe your sample? This goes also for many parts of your results that need to be backed with your research data, either as supplement or as an open source data base.

Thank you for noting this. We apologize that we missed updating this earlier with the appropriate published article. This has now been updated in the reference section.

Reviewer 3 Report

This is a well-written manuscript, highlighting the impact of COVID restrictions on women's physical activity levels. Some suggestions for the authors follow. 

Lines 19 - 20: Comma after the word restrictions to read: 'Given the challenges that women uniquely face due to restrictions, it is imperative to advocate and provide environmental opportunity and support...'

Lines 51-52: Are these factors cited as barriers to being physically active in general or during COVID, specifically? I think time constraints are often reported by women as barriers to being physically active, especially those with dependents, so I'm wondering if this is unique to the pandemic? Perhaps the authors could describe this point in terms of the increase relative to pre-COVID. Seeing statistics relative to pre-COVID (if available) could strengthen the point. 

Lines 83 - 88: Were all the questions asked relative to before COVID? On line 83, the authors report that participants were requested to report whether levels of physical activity had changed or remained the same. Is this true of the rest of the additional questions? Was there some sort of baseline measure? The study was carried out during the peak of COVID restrictions, so were the questions set out as 'compared to two months ago, rate your...'. A little more information on the way the questions were administered would be helpful here. 

Line 101: What was the GAD-7 cut-off used to determine the presence of anxiety? Seeing as there is a significant difference between women and men in their anxiety scores, it'd be helpful if the authors noted the cut-off used, e.g., GAD-7 scores > 10.  

Line 122: Curious to know why there was a reduction of hours in women if there were no sex differences in child care. 

Line 157: Why is the data unreported? Seems like it'd be important to report if the claim is based on the data. 

Lines 198 - 199: states that women who had changes in their work routine had more anxiety levels, which make sense. But further down at lines 203 - 205, it reads as though women who had changes in their work routine were more motivated to engage in physical activity. I know the point is to highlight the difference in motivation for physical activity, but this is very apparent. When you read the section, it initially comes across as two conflicting points. Perhaps the authors can make this clearer by rephrasing. Also, not sure what the significance of mentioning the difference in motivation adds to the findings. It's already been established that women who were already engaged in physical activity faired better than those who hadn't - COVID simply amplifies these differences. Women who were already inactive are no more inactive. 

Author Response

This is a well-written manuscript, highlighting the impact of COVID restrictions on women's physical activity levels. Some suggestions for the authors follow. 

Thank you for taking the time to review this manuscript.

Lines 19 - 20: Comma after the word restrictions to read: 'Given the challenges that women uniquely face due to restrictions, it is imperative to advocate and provide environmental opportunity and support...'

Thank you this has been updated.

Lines 51-52: Are these factors cited as barriers to being physically active in general or during COVID, specifically? I think time constraints are often reported by women as barriers to being physically active, especially those with dependents, so I'm wondering if this is unique to the pandemic? Perhaps the authors could describe this point in terms of the increase relative to pre-COVID. Seeing statistics relative to pre-COVID (if available) could strengthen the point. 

Thank you for this comment. We have added “in pre-pandemic studies” to ensure that the reader understands that this is referring to what we know about physical activity in women prior to COVID. The intention of this article was to assess how these barriers had changed since COVID in women so we hope that the reader finds an answer to this question after reading through our results. Given that we did not collect any data from study participants prior to the implementation of COVID-19 restrictions, we would be unable to directly assess changes to these barriers aside from the self-reported experiences of participants.

Lines 83 - 88: Were all the questions asked relative to before COVID? On line 83, the authors report that participants were requested to report whether levels of physical activity had changed or remained the same. Is this true of the rest of the additional questions? Was there some sort of baseline measure? The study was carried out during the peak of COVID restrictions, so were the questions set out as 'compared to two months ago, rate your...'. A little more information on the way the questions were administered would be helpful here. 

Thank you for pointing out this confusion. Each section has been updated to state “at the time of survey completion during COVID-19” to clarify that these were measures during COVID-19 AND not in comparison to prior to COVID-19.

Line 101: What was the GAD-7 cut-off used to determine the presence of anxiety? Seeing as there is a significant difference between women and men in their anxiety scores, it'd be helpful if the authors noted the cut-off used, e.g., GAD-7 scores > 10.  

Thank you. We have added this to the methods section which now has the following statement to clarify the confusion “The GAD-7 is based on seven items that are scored on a scale of 0 to 3 with a total possible score of 21. Cut off scores of 5, 10 and 15 were used as a score of mild, moderate and severe anxiety symptoms respectively.” (Kroenke et al., 2007).

Line 122: Curious to know why there was a reduction of hours in women if there were no sex differences in child care. 

Unfortunately we do not have specific data to answer this comment but we would assume that it is due to women holding more positions in the service industry which was more negatively affected by job losses during the pandemic. We wish we had asked additional questions related to childcare but at the time of study design we missed doing so.

Line 157: Why is the data unreported? Seems like it'd be important to report if the claim is based on the data. 

Thank you for this comment. We agree and have added this data to Table 5 in order to provide a clearer picture for the reader.

Lines 198 - 199: states that women who had changes in their work routine had more anxiety levels, which make sense. But further down at lines 203 - 205, it reads as though women who had changes in their work routine were more motivated to engage in physical activity. I know the point is to highlight the difference in motivation for physical activity, but this is very apparent. When you read the section, it initially comes across as two conflicting points. Perhaps the authors can make this clearer by rephrasing. Also, not sure what the significance of mentioning the difference in motivation adds to the findings. It's already been established that women who were already engaged in physical activity faired better than those who hadn't - COVID simply amplifies these differences. Women who were already inactive are no more inactive. 

Thank you for this comment. This comment is based on a misunderstanding of self-determination theory, as individuals who indicate higher levels of controlled motivation and introjected regulation are less self-determined, and therefore less motivated, than individuals who score high on autonomous or intrinsic motivation. The follow statement has been added to enhance clarity:

“These results indicate that women who experienced work-related changes experienced less self-determined motivation to engage in physical activity.”

Reviewer 4 Report

I am grateful for reviewing this paper. This paper is regorously well described and considered to contribute to the academic development of public health. This study is thought to be meaningful in that it investigated the physical activity and mental well-being level due to the recent pandemic COVID-19. However, this paper needs minor some revisions, as presented.

<Abstract>

  1. In line 14, ~ more generalized anxiety then men, replace ‘then’ with ‘than’.

< Materials and Methods>

  1. Please describe study design and the total number of study participants.
  2. Please describe how to the sampling framework of the national survey in Canadian population.
  3. In ethical considerations, Please describe the anonymity guarantee of personal information of collected data,
  4. Please describe by dividing into data analysis and research result areas.
  1. Describe how to data analysis, and additionally describe statistical techniques for securing representativeness of national survey, such as hierarchy (layer variables), cluster (collection variables), and sample weight (weight variables) for representative samples.
  2. In the results, describe only the research results based on tables or figures accurately and concisely.
  3. It is estimated that the number of participants by age groups among those over 19 years old may have an influence on the results of this study. Therefore, in demographic characteristics, please describe the number of subjects by age groups.
  4. In lines 75-77, As suggested in those sentences, “demographic characteristics included age, sex, marital status, occupational status (including changes due to COVID-19) and changes in childcare obligations (due to COVID-19)”, add the relevant demographic characteristics as a table.

< Discussion>

  1. In line 218, more generalized anxiety then men, replace ‘then’ with ‘than’.
  2. Atrophy of physical activity and the occurrence of psychosocial and mental health problems due to COVID-19 are of course expected problems. If so, this study must provide the evidences for the differecnes in sex, age groups, marital status, occupational status (including changes due to COVID-19) and changes in childcare obligations (due to COVID-19) etc.

Author Response

I am grateful for reviewing this paper. This paper is regorously well described and considered to contribute to the academic development of public health. This study is thought to be meaningful in that it investigated the physical activity and mental well-being level due to the recent pandemic COVID-19. However, this paper needs minor some revisions, as presented.

Thank you for taking the time to review this paper.

<Abstract>

In line 14, ~ more generalized anxiety then men, replace ‘then’ with ‘than’.

Thank you this has been updated.

< Materials and Methods>

Please describe study design and the total number of study participants.

Thank you this has been updated to state “ Study participants were Canadian men (n=215) and women (n=871) over the age of 19 who were recruited for a national study on physical activity and well-being during COVID-19. The original study has been described in detail elsewhere (Lesser and Nienhuis, 2020) and was a cross-sectional study design.”

Please describe how to the sampling framework of the national survey in Canadian population.

The intent of sampling was not necessarily to secure a full representative sample from the Canadian population. The expectant limited timeframe the COVID-19 restrictions were in place did not permit a more comprehensive recruitment strategy. Several limitations of the recruitment strategy are identified within concluding paragraphs of the manuscript as we did not use a targeted sampling method.

In ethical considerations, Please describe the anonymity guarantee of personal information of collected data,

This has been revised to state “all participants provided online informed consent and were ensured of anonymous data collection.”

Please describe by dividing into data analysis and research result areas.

We are not sure what this comment is directly asking but are hopeful that we have appeased your comment by including the a statistical analysis section as shown below:

“2.3. Statistical Analysis

Descriptive statistics of demographic characteristics were conducted, and independent t-test’s and chi square tests were conducted to compare demographic differences across sex. To analyze physical activity behaviour and well-being outcomes of women, participants were categorized based on changes to physical activity behaviour and subsequent comparative analysis was performed utilizing one-way ANOVA’s. Multiple one-way ANOVA’s were conducted to examine the number of moderate to vigorous physical activity minutes and barriers and facilitators to physical activity engagement. Bivariate analysis and independent sample t-tests were conducted to compare motivation levels between men and women, and additional t-tests were utilized to explore the impact of childcare changes on physical activity and well-being measures. SPSS-25.0 software was utilized to compute all statistical analysis and significance was set at p < .05. “

Describe how to data analysis, and additionally describe statistical techniques for securing representativeness of national survey, such as hierarchy (layer variables), cluster (collection variables), and sample weight (weight variables) for representative samples.

While this is a good comment, as stated in the earlier comment we did not set out to have a targeted sampling framework and have included this in the limitation section. However, we do believe that our large sample reduces the likelihood of a disparate population.

In the results, describe only the research results based on tables or figures accurately and concisely.

Thank you. Clear table references have been provided and corresponding paragraphs routinely discuss table contents.

It is estimated that the number of participants by age groups among those over 19 years old may have an influence on the results of this study. Therefore, in demographic characteristics, please describe the number of subjects by age groups.

Thank you. We agree and have added a new Table 1 has been created that summarizes demographic data.

In lines 75-77, As suggested in those sentences, “demographic characteristics included age, sex, marital status, occupational status (including changes due to COVID-19) and changes in childcare obligations (due to COVID-19)”, add the relevant demographic characteristics as a table.

Thank you. We agree and have added a new Table 1 has been created that summarizes demographic data.

< Discussion>

In line 218, more generalized anxiety then men, replace ‘then’ with ‘than’.

This has been updated.

Atrophy of physical activity and the occurrence of psychosocial and mental health problems due to COVID-19 are of course expected problems. If so, this study must provide the evidences for the differecnes in sex, age groups, marital status, occupational status (including changes due to COVID-19) and changes in childcare obligations (due to COVID-19) etc.

While we appreciate this comment it was not the objective of this paper to compare across different groups and we do not believe we have the statistical power to do so. While it is true that a reduction in physical activity and mental health challenges may have been expected during the pandemic the goal of scientific research is to provide data to support hypotheses. Additionally, given that this pandemic is unprecedented and novel in the scientific literature it is important to provide evidence for these changes in the case of a future challenge to society.